# Oral Squamous Cell Carcinoma Is Associated with a Low Thrombosis Risk Due to Storage Pool Deficiency in Platelets

**DOI:** 10.3390/biomedicines9030228

**Published:** 2021-02-24

**Authors:** Pierre Haen, Lydie Crescence, Diane Mege, Alexandre Altié, Christophe Dubois, Laurence Panicot-Dubois

**Affiliations:** 1Aix Marseille Université, INSERM 1263, INRAE, C2VN, 13885 Marseille, France; pierre.haen@yahoo.fr (P.H.); lydie.crescence@univ-amu.fr (L.C.); dr.dianemege@gmail.com (D.M.); alexandre.altie@univ-amu.fr (A.A.); christophe.dubois@univ-amu.fr (C.D.); 2Department of Oral and Maxillofacial Surgery, Laveran Military and Academic Hospital, 13384 Marseille, France; 3Aix Marseille Université, PIVMI, 13885 Marseille, France; 4Department of Digestive Surgery, Aix Marseille Univ, APHM, Timone University Hospital, 13885 Marseille, France

**Keywords:** disorders of platelet function, intravital microcopy, laser injury, oral squamous cancer, platelet granules

## Abstract

Venous thrombo-embolism (VTE) disease is the second most common cause of mortality in cancer patients, and evaluation and prevention of thrombosis risk is essential. VTE-associated risk varies according to the type of tumor disease. Oral cancer is the most frequent type of head and neck cancer, and it represents approximately 2.1% of all cancers worldwide. Most tumors are squamous cell carcinomas and are mainly due to tobacco and alcohol abuse. VTE risk associated with oral squamous cell carcinoma (OSCC) is low. However, many studies have shown that OSCC has the following biological features of cancers associated with a high thrombosis risk: modified thrombosis and fibrinolysis mechanisms; strong expression of procoagulant proteins; secretion of procoagulant microparticles; and production of procoagulant cytokines. Using an original mouse model of tongue squamous cell carcinoma, our study aimed to clarify this paradoxical situation. First, we showed that OSCC tumors have a pro-aggregatory phenotype and a high local thrombosis risk. Second, we found that tongue tumor mice do not have an elevated systemic thrombosis risk (the risk of an “at distance” thrombosis event such as lower extremity deep venous thrombosis or pulmonary embolism) and even show a reduction in risk. Third, we demonstrated that tongue tumor mice show a reduction in platelet reactivity, which explains the low systemic thrombosis risk. Finally, we found that tongue tumor mice present granule pool deficiency, thereby explaining the reduction in platelet reactivity and systemic thrombosis risk.

## 1. Introduction

The association between venous thrombo-embolism and malignancy was first described by Bouillaud and Trousseau [1,2] in the 19th century. Currently, it has been clearly shown that the association exists for a hypercoagulable state associated with malignancy and that thrombosis risk differs according to the cancer features (histological features, localization, and stages) and cancer treatments used (chemotherapy, surgery, and radiotherapy) [3].

Venous thrombo-embolism disease, especially pulmonary embolism, is the second most common cause of mortality in cancer patients. Thus, evaluation and prevention of thrombosis risk is essential [4]. Anticoagulant therapy is indicated in the prevention of venous thromboembolism (VTE), but its use must be limited in high-risk patients because it may also be responsible for complications, such as hemorrhage or immuno-allergic reactions [3].

Oral cancer is the most frequent type of head and neck cancer, and it represents approximately 2.1% of all cancers worldwide [5,6]. Anatomically, oral localization corresponds to the lip, tongue, hard palate, gingivae, and internal side of the cheek. Oral tongue cancer is the most frequent sublocalization [7]. Histologically, squamous cell carcinoma accounts for 90% of all malignancies in the oral cavity. Tobacco use and excessive alcohol consumption are the most common risk factors. Papillomavirus infection has also been shown to be a risk factor, especially for tongue cancer, for which epidemiology is thus slightly different (younger patients without alcohol or tobacco abuse) [8]. Thrombosis risk associated with oral squamous cell carcinoma (OSCC) has been empirically shown to be low or nonexistent [9,10]. For some authors, however, this risk is demonstrable [11,12], and many studies have shown that OSCC has the following biological features of cancers associated with a high thrombosis risk: modified thrombosis and fibrinolysis mechanisms [13,14]; strong expression of procoagulant proteins, such as factor tissue (FT) [15,16]; secretion of procoagulant microparticles [17]; and production of procoagulant cytokines [18]. As we have previously suggested in our review [19], a paradoxical situation exists for OSCC tumors, since these tumors present all the features of high thrombosis risk associated cancer but show a low associated systemic thrombosis risk. Our present study aimed to clarify this situation. First, our study highlighted this paradox by confirming the pro-thrombotic profile of OSCC tumors and by exploring the associated systemic venous thrombo-embolism (VTE) risk, which is the risk of a “at distance” thrombosis event such as lower extremity deep venous thrombosis or pulmonary embolism. Thereafter, we identified the potential mechanisms that may explain these facts using an original mouse model of tongue squamous cell carcinoma.

## 2. Materials and Methods

### 2.1. Antibodies and Reagent

Alexa 649 rat anti-mouse GPIb antibody (Emfret) was used in vitro for immunofluorescence microscopy and flow cytometry experiments, an in vivo for intravital microscopy. A homemade Alexa 488-conjugated mouse anti-fibrin antibody (the hybridoma was generously provided by Prof B. Furie) was used in vitro for flow cytometry experiments, an in vivo for intravital microscopy. Alexa 649 and Alexa 488 irrelevant antibodies respectively from Beckman and homemade, were used in vitro for immunofluorescence microscopy. PE Conjugated rat anti-mouse CD41/CD61 (activated form, JON/A Emfret, Eibelstadt, Germany) and microbeads (Biocytex, Marseille, France) were used in vitro for flow cytometry experiments. ADP (Sigma-Aldrich, St Quentin Fallavier, France) and TRAP (PolyPeptide, Strasbourg, France,) were used for platelet aggregation experiments.

### 2.2. Cell Culture

The AT-84 oral cancer cell line is derived from a murine tumor of the oral mucosa that spontaneously arose in a C3H mice, it was originally isolated by Hier et al. [20] and was kindly provided by Dr Venuti A. and Dr Paolini F. [21] (Regina Elena National Cancer Institute, Rome, Italy). AT-84 cells were grown in RPMI-1640 medium (Life Technologies, Courtaboeuf, France) supplemented with 10% FCS (PAA), 100 U/mL penicillin (Life Technologies), 100 mg/mL streptomycin (Life Technologies) and 0.1% fungizone (Life Technologies). Cells were grown at 37 °C in a humidified atmosphere with 5% CO_2_.

The Hight TF-Panc02 cell line was derived from a pancreatic ductal adenocarcinoma induced in a C57BL/6 mouse^1^. This cell line is a subclone of Panc02 cells transfected with pcDNA 6.2-GW/EmGFP-miR mock described previously by Mezouar et al. [22]. Hight TF-Panc02 were grown in RPMI-1640 medium (Life Technologies) supplemented with 10% FCS (PAA), 100 U/mL penicillin (Life Technologies), 100 mg/mL streptomycin (Life Technologies) and 0.1% fungizone (Life Technologies). The cells were grown at 37 °C in a humidified atmosphere with 5% CO_2_.

### 2.3. Mice

Wild-type C3H/HeNRj mice were obtained from Janvier Laboratories and were housed under standard conditions. All animal care and experimental procedures were performed as recommended by the European Community guidelines and were reviewed by the local ethical committee number 14 (number APAFIS#9330-201703017514353) and approved by the French ministry of research, education and innovation on the 06th December 2017.

### 2.4. TF and TFPI Activity Assay

A chromogenic assay (Actichrome TF and TFPI activity assay; Sekisui Diagnostics, Burlington, MA USA) was used to analyze TF and TFPI activity according the manufacturer’s instructions. We incubated 115,000 AT84 Cells (70,000 High TF PancO2 cells, control group) in a 96-well microplate for 24 h, then washed and added reagents. The absorbance was read at 405 nm with a microplate reader (MR5000; Dynatech, Nevada, USA). The TF activity and the TFPI concentration were determined by interpolation from a standard curve constructed using different amounts of lipidated TF and TFPI standards. To standardized results, TF activity and TFPI concentration values were recalculated on the basis of a single cell.

### 2.5. Mice Platelet Rich Plasma Preparation (PRP)

PRP were obtained from C3H/HeNRj mice under anesthesia (ketamine 125 mg/kg, xylazine 12.5 mg/kg, atropine 0.25 mg/kg). Blood was collected from the inferior vena cava on citrated tube. Citrated whole blood (1/9) was centrifuged at 90 g during 13 min. The upper phase was considered as platelet-rich plasma (PRP). Platelets were quantified by flow cytometry (Gallios, Villepinte, France) and adjusted to 125,000 platelet/μL with Tyrode when necessary.

### 2.6. Induction of Platelet Aggregation by OSCC Cell

AT-84 cells were dissociated using a non-enzymatic buffer and re-suspended in RPMI-medium not supplemented. Then RPMI-medium, platelet activator (ADP or TRAP) or different amounts of cells (final concentration 2 × 10^5^ cells/mL) were added to mice PRP and the extent of aggregation of platelets was measured by turbidimetric method using an aggregometer (APACT 4004, labitech, Ahrensburg, Germany).

### 2.7. Platelet Aggregometry

Platelet aggregation was tested in platelet rich plasma using a light transmission aggregometer (APACT 4004) under constant magnetic stirring at 37 °C. PRP samples were adjusted with Tyrode buffer to standardize platelet concentration in each sample (125,000 platelet/μL). Platelet stimulation was done using ADP (final concentration of 6 μM) or using thrombin receptor activating peptide (TRAP, final concentration of 150 μM). The concentration of each agonist was selected based on pilot studies that identified minimal concentrations that consistently resulted in approximately 70% aggregation of wild type mouse platelets (data not shown). Aggregation test time was 1200 s and stimulation occurred at the 100th second.

### 2.8. Mouse Model of Tongue Squamous Cell Carcinoma

AT-84 cells were cultured to 80% confluence. Once the cells reached the exponential growth phase, they were washed three times with PBS (Phosphate Buffered-Saline 1X, Thermo Fisher, Villebon sur Yvette, France) and briefly exposed to nonenzymatic cell dissociation buffer (Thermo Fisher) to dislodge the cells. Cells were carefully washed three times with PBS, resuspended in RPMI medium and diluted to the desired concentration. Ten-week-old C3H mice were anesthetized with isoflurane, and a tumor cell suspension (8 × 10^5^ AT84 cells in 30 μL of RPMI medium, or 30 μL of RPMI) were injected into the musculature of the mobile tongue. Buprenorphine was subcutaneously administered to mice (0.1 mg/kg) just after the surgery. Mice were observed daily and treated with buprenorphine if distinctive pain signs were detected. After eight days, mice lost approximately 15–20% of their weight and did not present any pain. At this time, mice had tongue tumors that were approximately a third of the mobile tongue. All experimentations of this study were performed at this stage.

### 2.9. Intravital Microscopy and Laser-Induced Injury

Mice were anesthetized, and a tracheal tube was inserted. A jugular vein cannula was inserted to infuse into blood circulation (anesthesia and antibodies). Cremaster preparation and intravital video microscopy were performed as previously described using a Slide-Book (Intelligent Imaging Innovation, London, UK) [23]. Vessel wall injury was induced with a MicroPoint Laser System (Photonics Instruments, Pittsfield, MA, USA) focused through the microscope’s objective, and the parfocal focused on the focal plane and aimed at the vessel wall as previously described [24]. The analyses were performed using SlideBook software as described previously by Dubois et al. [25].

### 2.10. Optical Microscopy

Mouse tongues were collected, formalin fixed and embedded in paraffin. Four micrometer paraffin embedded tumors sections were mounted on slides and stained by hematoxylin-eosin-safran (HES). Optical microscopy was performed using a Leica DMI8.

### 2.11. Fluorescent Microscopy

Mouse tongues were collected and stored in Optimal Cutting Temperature (OCT) at −80 °C. Four micrometer OCT glue-embedded tumors sections were mounted on slides, acetone fixed, protein saturated by Bovine Serum albumin and submitted to labeling by different antibodies. Tongue sections were analyzed in fluorescent microscopy using a Leica DMI8.

The fluorescence analysis was done using Fiji (ImageJ, Bethesda, MD, USA). Native fluorescent images were imported in the software, and a 1500 × 750 pixels sized rectangle located in the tumor area was analyzed. Sums of pixels’ intensity were calculated and compared.

### 2.12. Transmission Electron Microscopy (TEM)

TEM analysis was carried out by resuspending mice PRP to an equal volume of 2% glutaraldehyde in 0.1 M phosphate buffer, pH 7.4., overnight. Then samples were centrifuged at 1000× *g* for 10 min, washed, and post-fixed with 1% osmium tetroxide in 0.1 M phosphate buffer for 60 min at 4 °C. After that they were dehydrated through a graded ethanol series and embedded in resin. Ultrathin sections were prepared, stained with uranyl acetate and lead citrate, and examined under a transmission electron microscope (JEM 1400 JEOL, Tokyo, Japan).

### 2.13. Statistics

Significance was determined by the Mann–Whitney test for the in vivo experiments and Student t test for the in vitro experiments. Differences were considered significant at *p* < 0.05.

## 3. Results

### 3.1. OSCC Cells Induce Platelet Aggregation In Vitro

To investigate the ability of OSCC cells to induce platelet aggregation, we evaluated murine platelet aggregation (using PRP) after adding suspended AT-84 cells in serum free to RPMI. No aggregation was noted when platelets were incubated alone or with RPMI. At a threshold AT-84 cell density (4 × 10^5^ cells/mL, other concentrations not shown), significant platelet aggregation occurred on part with that induced by platelet agonist (high concentration of ADP, final concentration of 50 μM, Figure 1a,b). We observed with the AT84 cells a latent period between cell adjunction and platelet aggregation which has been reported by Chang et al. in a similar experiment using human tongue cancer cells. This latent period was dependent on the final OSCC cell concentration. As illustrated in Figure 1c, a doubling of the latency time occurred when the cell concentration was decreased by five (5 × 10^5^ cells/mL versus 1 × 10^5^ cells/mL). Chang et al. also reported that normal oral cells (e.g., fibroblast buccal and gingival epithelial cells) do not induce platelet aggregation [16].

Tissue factor (TF) is a key regulator of platelet aggregation and hemostasis in vivo. Even if many biological mechanisms of tumor cell-induced platelet aggregation exist [26], TF activity enhancement by tumor cells is considered as the predominant mechanism implicated in cancer-induced hypercoagulability [27]. Indeed, TF is the first molecule of the extrinsic coagulation cascade. The presence or expression of TF alone is not sufficient to exert an effect on platelet aggregation. TF must be in its active or decrypted form to initiate the coagulation cascade. Thus, the level of TF activity reflects the level of hypercoagulability of a cancer cell line [28]. Tissue factor pathway inhibitor (TFPI) is the specific inhibitor of TF and consequently the coagulation cascade and platelet aggregation. The balance between TF and TFPI activity supported by a cancer cell reflects the real risk of thrombosis associated with cancer cells [22,29]. Using TF and TFPI activity assays, we measured and compared the TF and TFPI activities of AT-84 cells to those of a mouse pancreatic cancer cell line (Hight TF Panc02 cells) which had elevated TF activity and low to moderate TFPI activity. [22]. AT-84 cells presented significantly higher TF activity (Figure 1d, *p* < 0.05) than did the Hight TF PancO2 cell line. Moreover, AT-84 cells presented significantly lower TFPI activity (Figure 1e, *p* < 0.001) than did the Hight TF PancO2 cells [22]. Taken together, these results highlighted the strong ability of AT-84 cells to induce platelet aggregation and confirmed their pro-aggregatory phenotype.

### 3.2. Platelet Aggregates and Fibrin Deposits Are Locally Detected at the Site of the Primary Squamous Cell Carcinoma Tumors

After confirmation of the pro-aggregatory phenotype of OSCC cells in vitro, we investigated this phenomenon in vivo by measuring platelet aggregation and fibrin generation (final product of aggregation) in tongue tumors. We used our mouse model of tongue OSCC generated by intralingual AT-84 cell injection. After eight days of tumor growth (Figure 2a right), mice were sacrificed, and tongues were collected, fixed, and sliced. Slides were stained using standard dyes (hematoxylin, eosin, and safranin) and observed under an optical microscope. Optical examination (Figure 2a middle panel) revealed spindle-shaped cells with a pronounced storiform or sarcomatoid pattern. Nuclei were elongated into an oval shape, and they were plump with multiple chromocenters and often exhibited one or more prominent eosinophilic nucleoli. The cytoplasm was moderately abundant, slightly granular and lightly eosinophilic (Figure 2a right panel). Some necrotic areas were also observed. Histological features confirmed the presence of tongue squamous cell carcinoma (SCC) and more specifically sarcomatoid carcinoma. 

Another set of slides was incubated with an Alexa 649-conjugated rat anti-mouse CD42b antibody to highlight platelet aggregates and an Alexa 488-conjugated mouse anti-fibrin antibody to highlight fibrin deposits. Similar procedures were also performed on control mouse tongues as well as performed with nonspecific antibodies to confirm the specificity of our results.

Fluorescence microscopy (Figure 2b,c) showed cluster of high fluorescence intensity clusters in tumor areas indicative of the presence of platelets and fibrin deposits, which were not observed in the same regions of tongues without tumors. Fiji analysis showed a significant elevation in the fluorescence intensity in tongues with tumors compared to tongues without tumors (Figure 2d,e). The fluorescence intensity of slides stained with nonspecific antibodies was low and was from the platelets and fibrin fluorescence intensity thus validating the specificity of the findings. Taken together, these data confirmed the existence of platelet and fibrin deposits at the site of primary SCC tumors, and they confirmed the ability of AT84 cells to induce platelet aggregation in vivo.

### 3.3. Tongue SCC Is Associated with a Reduction in Systemic Thrombosis Risk in Living Mice

After demonstrating the pro-aggregatory phenotype of AT84 cells in vitro and in vivo, we next explored the systemic thrombotic risk associated with AT84 tongue tumors in vivo. Systemic thrombotic risk was considered the general risk of occurrence of a thrombotic event as a lower extremity deep venous thrombosis (phlebitis) or pulmonary embolism, which are always known to have a high incidence in malignancies. First, we examined whole blood samples to best characterize our mouse model of tongue tumors, especially to ensure that there were no issues with the number of blood cells potentially implicated in hemostasis. Whole blood examination was performed by flow cytometry, which did not identify any difference between mice with and mice without tongue tumors (Table 1). Moreover, we measured the tail bleeding time to test primary hemostasis. Although there was a trend of slightly elevated bleeding times in mice with tongue tumors, no significant difference was identified between the two groups (Table 1). These results were in accordance with the clinical observation of mice, which did not present any spontaneous sign of hemostasis perturbation (ecchymosis, hematoma, or abnormal bleeding). To further evaluate the systemic thrombotic risk associated with OSCC, we investigated thrombosis using real-time intravital microscopy protocol. We examined the kinetics of arteriolar thrombus development (platelet accumulation and fibrin generation) in real-time in the cremaster microcirculation of mice with and mice without tongue tumors. Thrombus formation was initiated by laser-induced injury of the arteriolar vessel wall. In tumor-free mice, platelets adhered to and accumulated at the site of injury (Figure 3a upper panel, Figure 3b right panel). The thrombus rapidly increased in size between 15 and 30 s after injury. The thrombus size then slightly decreased before increasing to its maximal value between 100 and 150 s after injury, and the thrombus then stabilized and decreased until the end of the measurement period. In mice with tongue tumors, platelets first accumulated at the site of injury (at approximately the same time between 15 and 30 s after injury), but the thrombus size quickly decreased and stayed low to the end of the experiment. (Figure 3a upper panel, Figure 3b right panel). Comparative analysis of the median area under the curve of the integrated fluorescent intensity and maximum integrated fluorescent intensity highlighted a significant difference (*p* < 0.001) with a decrease in platelet accumulation (Figure 3b, middle and right panel) between mice bearing tongue tumors and tumor-free mice.

In the tumor-free mice, fibrin generation started almost immediately following laser injury and increased with a nearly consistent slope to the end of the recording period (Figure 3a,c), which was similar to previous reports [22,30]. With regard to fibrin generation, in the tongue tumor group, the slope obtained was shallower than that obtained from tumor-free mice (Figure 3a,c). Comparative analysis of the median area under the curve of integrated fluorescent intensity and maximum integrated fluorescent intensity highlighted a significant difference (*p* < 0.001) with a decrease in platelet accumulation and fibrin generation in mice with tongue tumors (Figure 3c, middle and right panel). These data indicated a reduction in the systemic thrombosis-associated risk in mice with tongue SCC mice using this vessel injury model. Taken together, our results indicated that AT84 cells have a pro-aggregatory phenotype in vitro and locally in vivo but that the systemic thrombotic risk associated with tongue tumors is not present in our mouse model. Moreover, this risk seems to be lower in mice with tongue tumors mice than in tumor-free mice according to the real-time thrombosis study protocol.

### 3.4. Tongue Tumors Are Associated with a Decrease in Platelet Reactivity

In our model of intravital microscopy with vessel laser injury, the thrombosis mechanism is thrombin dependent [31]. Briefly, laser injury induces endothelial cell activation and neutrophil recruitment (interaction of ICAM 1/LFA1). Neutrophils are the first cells to be recruited at the site of injury and are the main source of tissue factor (TF) in the laser injury model [30]. TF activates the coagulation cascade and induces thrombin generation, platelet activation (depending on thrombin and ADP release from platelets) and so thrombus formation [32]. Fibrin formation is due to coagulation activation and is independent of platelet accumulation [33]. Therefore, we investigated the influence of neutrophil accumulation in thrombus formation. We were not able to detect any difference in neutrophil accumulation at the site of injury in mice with tongue tumors compared to tumor-free mice (Median of area under curve of integrated fluorescence intensity of neutrophils, tumor-free mice: 2.9 × 10^−9^ AU, tongue tumor mice: 3.6 × 10^−9^ AU; *p* = 0.2278). Moreover, we verified that the blood neutrophil and platelet concentrations in mice with tongue tumors were not significantly different from those in tumor-free animals (Table 1). Then, we focused on platelet functions, including activation and aggregation abilities. With PRP, we studied platelet aggregation in mice with tongue tumors using two different major activators implicated in our laser injury model as already published and validated by our team, namely thrombin and ADP [24,25,32]. Thrombin and ADP are also the main agonists involved in thrombosis associated with tumor [2]. We observed hypo-aggregability of platelets from mice with tongue tumors compared to tumor free mice. A significant reduction in platelet aggregation was found in mice with tongue tumors following activation by ADP (6 μM) and following activation by a low concentration of TRAP (75 μM) (Figure 4a,b). In contrast, no difference in platelet aggregation was observed with activation by a high concentration of TRAP (300 μM) (Figure 4c). To the same purpose, we detected the activated form of αIIbβ3 in platelets issued from mice with tongue tumors and tumor-free mice before and after activation by a high concentration of ADP (50 mM) or low concentration of TRAP (75 μM). In the presence of each agonist, the platelets from mice with tongue tumors presented a decrease in αIIbβ3 activation on their surface compared to that in platelets from tumor-free mice in presence of each agonist (Figure 4d). This decrease in platelet reactivity in mice with tongue tumors may explain the reduction in the general thrombosis risk in the laser injury model.

### 3.5. Tongue Tumors Are Associated with Granule Pool Deficiency

Tongue SCC is associated with a low risk of systemic thrombosis due to the reduction in platelet reactivity after an ADP and TRAP stimulation, with a small increase in activated αIIbβ3. The main source of ADP in platelets is dense granules [34]. This ADP pool is necessary to achieve a full platelet aggregation. We hypothesized that platelets from lingual tumor-bearing mice could present morphologic differences compared to platelets issued from tumor-free mice. Then, we studied the morphology of platelets of mice bearing tongue tumors using transmission electron microscopy (TEM). Platelets from the mice with tongue tumors were rounder in shape than those from tumor-free mice. These observations suggested that the granules in the platelets from mice with tongue tumors were modified (Figure 5a). Platelet delta-granules or dense granules contain mostly small molecules, including calcium, adenine nucleotides (ATP and ADP), αIIbβ3, serotonin, and pyrophosphate. We counted the number of dense granules in (sliced) platelets from mice with tongue tumors and tumor-free mice. A significant (*p* < 0.0001) decrease in number of dense and alpha granules was found in platelets from mice with tongue tumors (mean of 0.48 ± 0.03 dense granules and 4.4 ± 0.13 alpha granules per platelet) compared to platelets from tumor-free mice (mean of 1.6 ± 0.05 dense granules and 5.3 ± 0.15 alpha granules per platelet) (Figure 5b).

Thus, these findings suggest a deficiency of granule storage in mice with tongue tumors resulting in a lower activibility of platelets in bearing lingual tumor mice, independent of the agonist used. This could lead to a lower level of activated αIIbβ3 at the platelet surface.

Higher magnification TEM images highlighted the following ultrastructural modifications in platelets from mice tongue tumors. Platelets had a rounded shape with many large empty vacuoles. These TEM findings indicated that the deficiency in the granule storage pool may explain the decreased platelet reactivity and the decreased risk of systemic thrombosis highlighted in the laser injury model.

## 4. Discussion

Thrombosis risk associated with malignancy has been extensively studied for specific cancers (colon, lung, and pancreas), but thrombosis risk for OSCC has been poorly studied [19]. To the best of our knowledge, our study is the first to specifically explore the risk of thrombosis in tongue SCC using a live animal model. Malignancy is usually associated with elevated thrombosis risk, but the thrombosis risk associated with OSCC is low or even nonexistent [35]. In the literature, few nonsolid tumors, such as leukemia, have been reported to be associated with a poor thrombosis risk [36], or to deficiency in the dense granule pool (chronic lymphocytic leukemia) [37]. However, solid tumors are usually associated with an elevated risk of the thrombosis [3].

Even so, few studies have reported that a solid cancer is associated with a low specific risk of thrombosis, due to detailed associated mechanisms, such as acquired hemophilia [38] or acquired von Willebrand disease [39]. Our study indicates that the lack of thrombosis risk in a solid cancer could be due to a deficiency in the storage pool.

In the clinic, patients with cancer usually benefit from anticoagulation medication, especially with surgery or chemotherapy, which would enhance the thrombosis risk [3]. For OSCC patients and some head and neck cancer patients, recommendations for anticoagulation medication are unclear. Some authors propose a systematic anticoagulation therapy [40], whereas others limit this approach to patients with high risk of thrombosis due to other comorbidities [41]. These treatments are not without risk (hemorrhage and immune/allergic response), and their use must be considered thoroughly. Our study provides additional arguments to limit the administration of anticoagulant therapy in OSCC cancer patients.

OSCC is even more unique than originally thought because we showed that OSCC tumors had a pro-aggregatory phenotype, with a high risk of local thrombosis without enhancement of the risk of systemic thrombosis. AT84 cells had characteristics of cancer cells with high risk of thrombosis, and they induced platelet aggregation. In addition, AT84 cells presented high tissue factor activity and low tissue factor pathway inhibitor activity. Moreover, signs of coagulation activation were observed in OSCC tumors with platelet aggregation and fibrin deposits. All of these features are generally found in cancer associated with high risk of thrombosis [42]. However, this was not the case for our model for which systemic thrombosis risk was not enhanced due to decreased platelet reactivity with delta-storage pool deficiency, which counteracted the pro-aggregatory characteristics of OSCC. Despite this reduction in platelet reactivity, the risk of local thrombosis remained high in OSCC tumors because there was high local expression of procoagulant proteins, resulting in platelet aggregation. Moreover, the platelets from mice with tongue tumors maintained normal aggregation abilities in response to a strong activation resulting in a high concentration of TRAP.

This paradoxical situation may seem surprising, but it reflects observations in the clinic. Similar to the murine OSCC cell line in the present study, human OSCC cells have clearly established pro-aggregatory phenotype in vivo and in vitro. OSCC tumors are implicated in local thrombosis events, especially internal jugular vein thrombosis [43] and even carotid artery thrombosis [44]. However, no augmentation of the systemic thrombosis risk occurs in humans with OSCC [19], suggesting that a mechanism to counteract the thrombotic potential of OSCC exists in humans. This mechanism is actually unknown in human OSCC, and we hypothesize that it could be the same mechanism as the decreased platelet reactivity observed in our animal model. Several studies have reported decreased platelet reactivity in cancer patients, notably hematological cancer [45] and ovarian cancer [46] patients. One could postulate that the presence of cancer cells in the bloodstream induced exhausted platelets which can be encountered in cancerous patients even in the absence of consumption coagulopathy. However, we reported that OSCC tumors express high levels of active TF, inducing a Tumor Cell Induced Platelet Aggregation (TCIPA) when present in blood. Alternatively, tumors could secrete ADP [2]. ADP may in turn activate platelets leading to the desensibilization of the P2Y1 receptor and the secretion of the granule contents, leading to exhausted platelets insensitive to ADP. In this case, activated and not resting platelets will be observed by electron microscopy. This hypothesis does not however explain the decrease in the reduced level of activated αIIbβ3 observed. Another possibility is that OSCC tumors will directly or indirectly act on megakaryocytes leading to the generation of deficient platelets.

Delta-storage pool deficiency is a thrombopathy defined by a reduction in granule secretion from platelets, which may be due to a quantitative defect (with a decreased number of granules per platelet) or a qualitative defect (abnormal content or abnormal mobilization of dense platelet granules). These defects lead to diminished platelet aggregation owing to the lack of a second wave of aggregation dependent on the release of dense granule content, especially ADP [47]. This phenomenon was observed in our mouse model of tongue SCC with a lack of platelet aggregation in the intravital real-time thrombosis study and with a decrease in platelet aggregation via activation of ADP or a low concentration of TRAP. Clinically, delta-storage pool deficiency manifests itself as hemorrhagic symptoms that are most of time moderate and often only observed following traumatism or a surgery, potentially without increasing the bleeding time [37]. In our animal study, the mice did not present a hemorrhagic profile, and we did not observe a significant elongation in the bleeding time. Thus, our results were not discordant with a delta-storage pool deficiency. The etiology of delta-storage pool deficiency is complex. This deficiency can be congenital or inherited, and it can be either isolated to a specific location or as part of the whole organism response. Delta-storage pool deficiency can also be acquired as has been observed in leukemia and myelodysplastic syndromes [36] as well as in inflammatory diseases, such as rheumatoid arthritis, lupus erythematosus [48] or postural orthostatic tachycardia syndrome [49]. To the best of our knowledge, this is the first report on the association between OSCC and delta-storage pool deficiency as well as the first report on this platelet disorder in a solid cancer. We also reported that this disorder may counteract with the prothrombotic profile of OSCC tumors and protect against a systemic thrombotic event. Even if a genetic disorder relating cellular transporters has been identified in an inherited form, the mechanism involved in acquired delta-storage pool deficiency remains unclear. It has been reported that failure of megakaryocyte maturation [50] occurs in the acquired form, which may be notable due to defective zinc homeostasis [51]. This approach would be interesting because a lack of zinc homeostasis has been reported in some cancers, especially head and neck cancer [52,53].

In conclusion, our mouse model of tongue tumors confirmed that OSCC tumors have a pro-aggregatory phenotype and may be involved in local thrombosis (in the tumor or peritumor area). However, we demonstrated that there was no elevated systemic thrombosis risk in a laser model. Therefore, we concluded that OSCC tumors are the only solid cancer that does not show an elevated risk of systemic thrombosis in the laser model. Moreover, we demonstrated that a delta-storage pool deficiency in mice with tongue tumors could be responsible almost partially for the decreased platelet reactivity, which counteracted the pro-aggregatory phenotype of OSCC tumors and explains the low risk of systemic thrombosis. This study provides a supplemental argument for limiting the systemic use of anticoagulant therapy in OSCC patients. The next step will be to identify the molecular mechanisms involved in delta-storage pool deficiency and to explore the distance relationship between OSCC tumors and platelets or megakaryocytes to deeply understand its role in the pathogenicity of the disease.

## Figures and Tables

**Figure 1 biomedicines-09-00228-f001:**
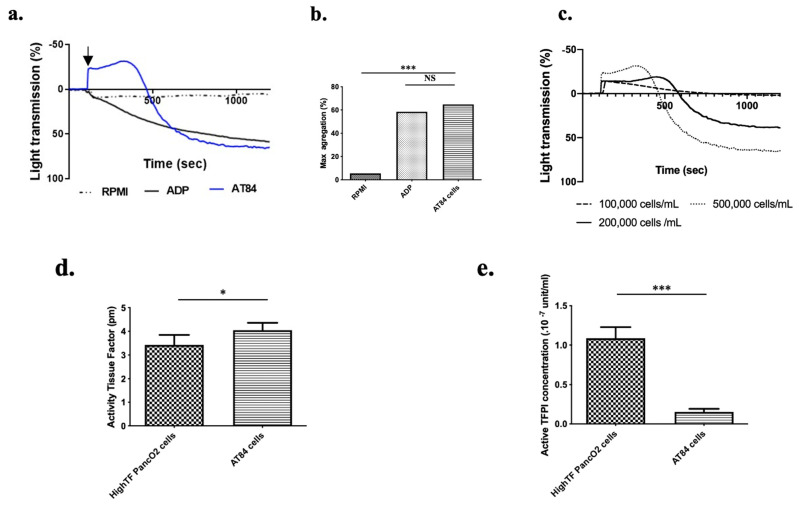
Oral squamous cell carcinoma (OSCC) cells induce platelet aggregation in vitro. (**a**) Mice showed typical platelet aggregation following injection of AT84 cells suspended in RPMI (black arrow) in PRP using a light transmission aggregometer (AT84 final concentration: 5 × 10^5^ cells/mL in PRP; platelet concentration: 1.25 × 10^8^ platelet/mL). Platelet aggregation induced by ADP (final concentration: 50 μM) was used as a positive control, and RPMI alone was used as a negative control. (**b**) Mean maximum platelet aggregation at 1200 s (*n* = 3 mice; 3 independent experimentations). (**c**) representative graph of lag time (s) as a function of AT84 cells concentration added to the reaction mixture. (**d**) Mean tissue factor activity (pM) and (**e**) tissue factor pathway inhibitor concentration (×10^−7^ unit/mL) per cell for AT84 and High-TF PancO2 cells (*n* = 4 independent experimentations). The results are expressed as the mean (bar) +/− standard error of mean (error bar). NS = nonsignificant. * *p* < 0.05 and *** *p* < 0.001.

**Figure 2 biomedicines-09-00228-f002:**
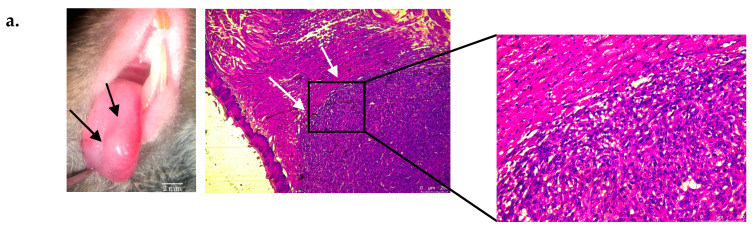
Localization of platelet aggregates and fibrin deposits in tongue squamous cell carcinoma (SCC) in vivo. (**a**, left panel) Macroscopic (optical magnification ×10) view of tongue squamous cell carcinoma eight days following intra lingual AT84 cell injection in C3H mice. Black arrows show tumor. (**a**, middle and right panel) Microscopic view (optical magnification × 50: middle panel) of a mouse tongue section with SCC (white arrows) with hematoxylin eosin safran staining. The right panel represent an enlargement (optical magnification × 200) of a mouse tongue section with SCC (**b**) Fluorescence microscopic imaging of mouse tongue section without (left side) and with (right side) SCC stained with platelet fluorescently labeled antibody (red) (optical magnification × 20 (upper pictures) and × 200 (lower pictures)). (**c**) Same views with fibrin staining (green). (**d**) Histogram showing results of platelet fluorescence analysis using Fiji (ImageJ, Bethesda) of a 1500 × 750 pixel sized rectangle located in the tumor area (magnification × 200). The results represent the sum of pixel intensity using platelet antibodies for tumor-free mice and mice with tongue tumors (*n* = 3 mice in each group with one pixel intensity measurement per mouse). (**e**) Histogram showing results of fibrin fluorescence analysis (*n* = 3 mice in each group with one pixel intensity measurement per mouse). The results are expressed as the mean (bar) +/− standard error of mean (error bar). * *p* < 0.05, *** *p* < 0.001.

**Figure 3 biomedicines-09-00228-f003:**
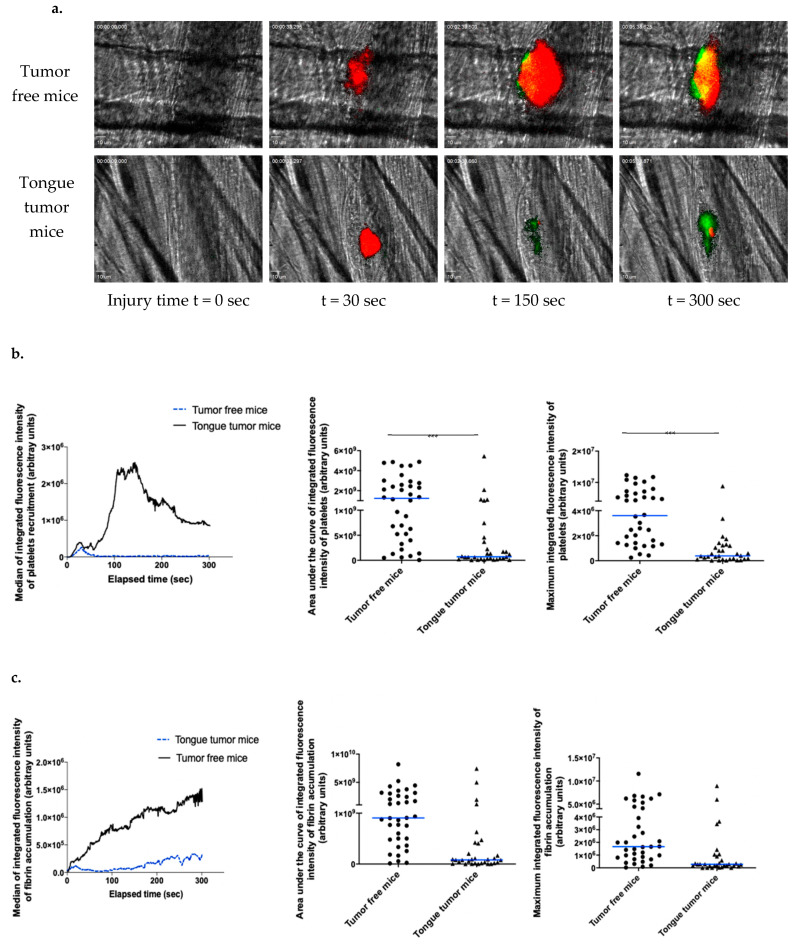
Kinetics of arterial thrombosis associated risk with tongue squamous cell carcinoma (SCC) in living mice. (**a**) Representative images of thrombus formation obtained by intravital microscopy in tumor-free mice and tongue tumor mice. Following laser-induced injury, the kinetics of thrombosis were evaluated based on the platelet (depicted in red) and fibrin (depicted in green) accumulation at different time points during thrombus formation by infusion of an anti-mouse platelet antibody and anti-fibrin antibody. (**b**) Graph on the left depicts the median integrated fluorescence intensity of platelets as a function of time in thrombi after laser-induced injury in tumor-free mice (37 thrombi in 3 mice) and tongue tumor mice (32 thrombi in 3 mice). Graph in the middle shows the area under the curve of integrated fluorescence intensity of platelets (each point corresponds to a thrombus) in both groups of mice. Graph on the right shows the median of the maximum integrated fluorescence intensity of platelets for both groups. (**c**) Each graph depicts the same results for fibrin accumulation. The results are expressed as median (bar in blue). *** *p* < 0.001.

**Figure 4 biomedicines-09-00228-f004:**
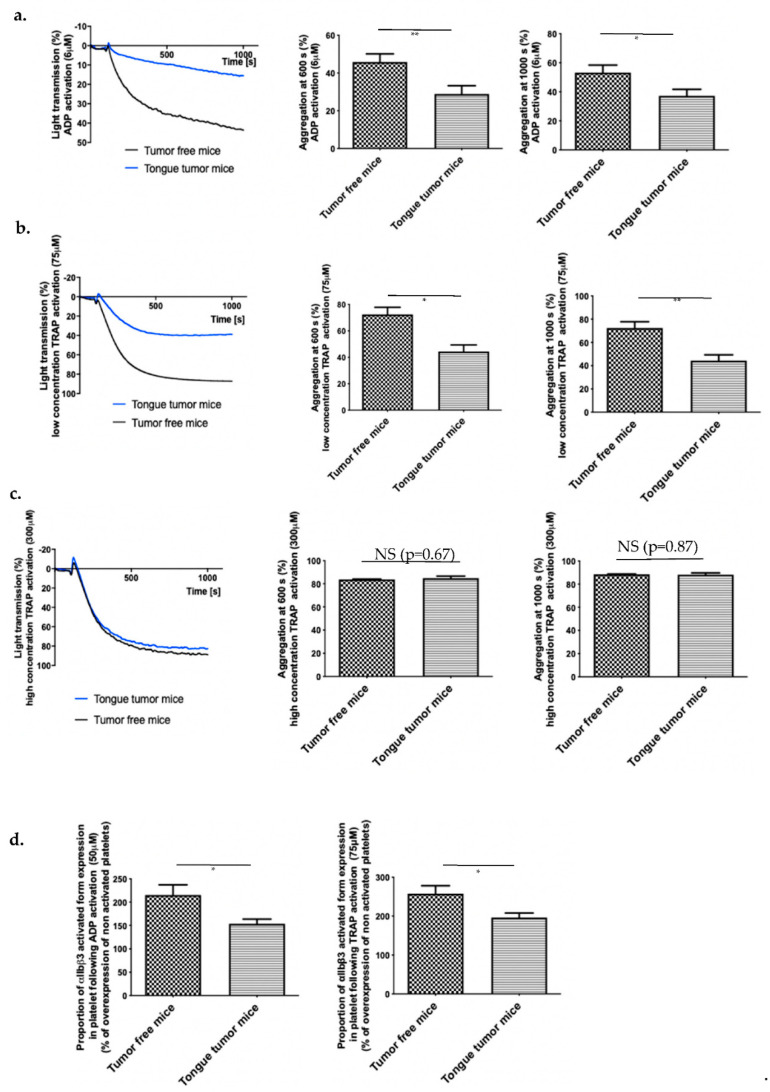
Platelet reactivity in mouse model of tongue squamous cell carcinoma (SCC). (**a**) Tumor-free mice (WT) and tongue tumor mice (AT84) platelet aggregation following treatment (at 100 s) with ADP (final concentration of 6 μM) in a light transmission aggregometer. Graphs in the middle and on the right side depict difference of platelet aggregation between tumor-free mice and tongue SCC mice at 600 and 1000 s following ADP activation (*n* = 8 mice in each group). (**b**) Same experimentation with low concentration (75 μM) TRAP activation (*n* = 5 mice in each group) and (**c**) high concentration (300 μM) TRAP activation (*n* = 5 mice in each group). (**d**) Proportion of platelets expressing activated αIIbβ3 following ADP (50 μM, on the left) or TRAP (75 μM, on the right) activation. Data represent the percentage compared to the expression before activation. The results are expressed as the mean (bar) +/− standard error of mean (error bar). * *p* < 0.05, ** *p* < 0.01, and NS: nonsignificant.

**Figure 5 biomedicines-09-00228-f005:**
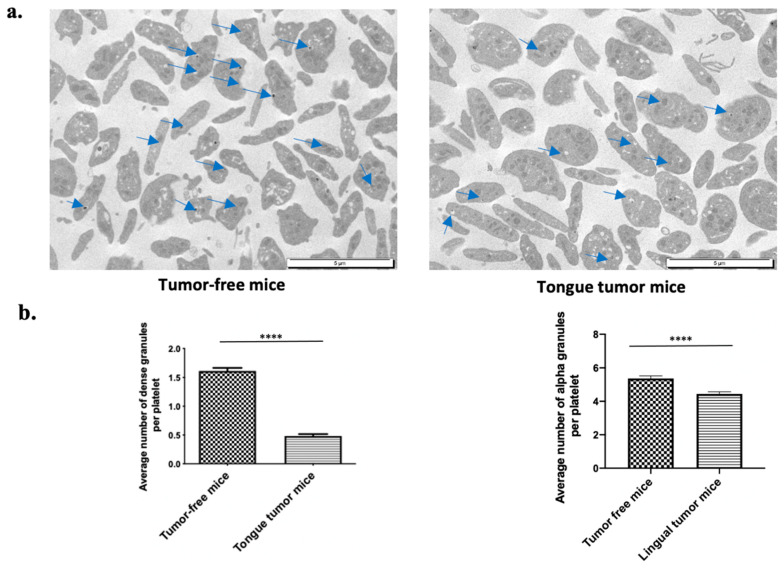
Morphological analysis of tongue tumor mouse platelets. (**a**) Transmission electron microscopy (TEM) examination of platelet (PRP) at 8000-fold magnification from tumor-free mice on left side and tongue tumor mice on right side. Blue arrows indicate granules. (**b**) Graphs depict the average number of dense granules per platelet (left panel) and the average number of alpha granules per platelet (right panel) for tumor-free mice and tongue tumor mice (*n* = 4 mice in each group; mean was calculated per 100 counted platelets; mean (bar) +/− standard error of mean (error bar); **** *p* < 0.0001). (**c**) TEM examination of platelet (PRP) at 20,000-fold magnification from tumor-free mice on left side and mice with tongue tumors on right side.

**Table 1 biomedicines-09-00228-t001:** Whole blood cells and tail bleeding time examination. Platelet, neutrophil, monocyte, and lymphocyte concentrations in whole blood for tongue tumor and tumor-free mice. Examinations were performed using flow cytometry (*n* = 12 in each group). Tail bleeding time measurement for tongue tumor and tumor-free mice (*n* = 10 in each group). SD: standard deviation, NS: non significant.

	Tumor-Free Mice (WT) Mean (SD)	Tongue Tumor Mice (AT84) Mean (SD)	P (Wilcoxon)
Platelets (nb × 10^3^/μL)	883.2 (399.6)	846.5 (254)	0.893 (NS)
Neutrophils (nb/μL)	1238 (486)	1381 (479)	0.588 (NS)
Monocytes (nb/μL)	159.8 (104.8)	242.7 (185.5)	0.168 (NS)
Lymphocytes (nb/μL)	3095 (1851)	2682 (1571)	0.685 (NS)
Tail bleeding time (seconds)	128.9 (28.48)	146.2 (48.62)	0.361 (NS)

## Data Availability

Original data could be asked to Laurence Panicot-Dubois.

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
