# Peer review of "Oral Squamous Cell Carcinoma Is Associated with a Low Thrombosis Risk Due to Storage Pool Deficiency in Platelets"

_biomedicines, 2021, doi:10.3390/biomedicines9030228_

Round 1

Reviewer 1 Report

Haen et al. report here on the prothrombotic and procoagulant status of oral squamous cell carcinoma. Using AT-84 cells the authors show that those cells induce platelet aggregation in vitro and present TF activity associated with low TFPI activity. Using a mouse model of tongue oral squamous cell carcinoma (OSCC) generated by intralingual AT-84 cell injection, the authors show the presence of squamous cell carcinoma (SCC), platelet (anti-CD42b), and fibrin (anti-fibrin) inside the tumor.

Then, using the tongue OSCC mice model, the authors investigated thrombus formation in vivo (laser-induced injury of the arteriolar vessel wall) and platelet aggregation in vitro (PRP stimulated with TRAP or ADP). Results show that both, thrombus size and platelet aggregation, were affected compared to control. The authors also identified an abnormal platelet morphology and a reduced number of dense granules in platelets from OSCC mice, and suggest that a defect in ADP release may explain the decrease of platelet response.

The findings are interesting and potentially useful for patient management. However, some concerns still need to be addressed to strengthen the conclusion.

1- The first set of experiments (figures 1 and 2) explore the ability of AT-84 cells to induce platelet aggregation in vitro and in vivo. However, this characterization is incomplete and should be improved. It would be interesting to know how AT-84 cells induce platelet aggregation in vitro: Is there a direct interaction between platelets and AT-64 cells or is the cells' secretion/supernatant important? What is TF implication in this mechanism?

In figure 1, the authors conclude that the lag phase between cell adjunction and platelet aggregation depends on cell concentration (not shown), these results should be shown. To thoroughly appreciate the cellular architecture, the authors should show a high magnification/resolution image of the mouse tongue section with SCC (figure 2a and comment lines 208-213).

2- Figures 4 and 5: The conclusion that defect in platelet aggregation is due to diminished ADP secretion is not substantiated by the data. The authors should show that addition of ADP restores platelet aggregation induced by low doses of epinephrine or collagen (CRP), two agonists known to be dependent on ADP secretion. If a lack of ADP is involved, aggregation should be restored to a level similar to the one observed with control platelets.

How do the authors explain that aggregation induced by ADP alone is affected?

3- An interesting point of this study is the defect of dense granules and whether the altered granule content is the reason behind the abnormality. However, more data to get insights into alpha and dense granules defects should be provided. For example, platelet vWF, fibrinogen and serotonin global content should be quantified by immunoassay/western blot or imaging. Dense granule release (ATP or serotonin release), as well as the α-granule release (PF4 or vWF release; ELISA assay) should be assessed. The number of α-granules should also be evaluated by EM and shown.

- Discussion: “Our study describes the first case of solid cancer lacking thrombosis risk due to a deficiency in the delta-storage pool.”: As such, the model is not fully supported by data and a more balanced take home message should be provided.

4- Another possible scenario is that carcinoma cells induced exhausted platelets which can be encountered in cancerous patients even in the absence of consumption coagulopathy. Assessing soluble activation markers (sCD40 or GPVI shedding) and the expression level of P-selectin would be valuable. This alternative model should be probed experimentally or at the very least be considered and discussed.

Minor comments:

1- Do the authors have any idea about the amount of cancer cell-derived microparticles in their mice ?

2- Line 308 (and line 319): TRAP is a synthetic peptide, the authors probably meant Thrombin?

3- The text size in the figures must be increased.

Author Response

Reviewer 1

Comments and Suggestions for Authors

Haen et al. report here on the prothrombotic and procoagulant status of oral squamous cell carcinoma. Using AT-84 cells the authors show that those cells induce platelet aggregation in vitro and present TF activity associated with low TFPI activity. Using a mouse model of tongue oral squamous cell carcinoma (OSCC) generated by intralingual AT-84 cell injection, the authors show the presence of squamous cell carcinoma (SCC), platelet (anti-CD42b), and fibrin (anti-fibrin) inside the tumor.

Then, using the tongue OSCC mice model, the authors investigated thrombus formation in vivo (laser-induced injury of the arteriolar vessel wall) and platelet aggregation in vitro (PRP stimulated with TRAP or ADP). Results show that both, thrombus size and platelet aggregation, were affected compared to control. The authors also identified an abnormal platelet morphology and a reduced number of dense granules in platelets from OSCC mice, and suggest that a defect in ADP release may explain the decrease of platelet response.

The findings are interesting and potentially useful for patient management. However, some concerns still need to be addressed to strengthen the conclusion.

1. The first set of experiments (figures 1 and 2) explore the ability of AT-84 cells to induce platelet aggregation in vitro and in vivo. However, this characterization is incomplete and should be improved. It would be interesting to know how AT-84 cells induce platelet aggregation in vitro: Is there a direct interaction between platelets and AT-64 cells or is the cells' secretion/supernatant important? What is TF implication in this mechanism?

We thank the reviewer for this comment.  We previously described, when cancer cells express high levels of TF and low levels of TFPI, that platelets are mainly activated through the tissue factor activity (Thomas et al JCI 2009, Palacios-Acedo et al frontiers on Immunology 2020, Mege et al Semin thromb  Hemost 2020). Activated TF coming from cancer cells or their microvesicles generates thrombin and induced the TCIPA (Tumor-cells Induced Platelet Aggregation; Mezouar et al IJC 2015 and Plantureux et al cancer 2019). To determine the potential role of microvesicles here, we determined the quantity of tumor microparticles secreted by AT84 cells.  Surprisingly, we observed using flow cytometry that AT84 cells secrete only 1,3 MPs/cell (+/- 0,2), comparing to the 32 MPs/cell (+/- 8) expressed by the mouse pancreatic cancer cell line Panc02. Also, we observed that the supernatant coming from AT84 cells did not induce the TCIPA. However, since the main goal our study was not to determine the role played by MPs on platelet aggregation, we did not add these results in the revised paper.

In figure 1, the authors conclude that the lag phase between cell adjunction and platelet aggregation depends on cell concentration (not shown), these results should be shown. To thoroughly appreciate the cellular architecture, the authors should show a high magnification/resolution image of the mouse tongue section with SCC (figure 2a and comment lines 208-213).

We modified the figure 2 according the reviewer comment and added a panel with a higher magnification (x200). Page 6 line 215, of the revised manuscript:

“Optical examination (figure 2a middle panel) revealed spindle-shaped cells with a pronounced storiform or sarcomatoid pattern. Nuclei were elongated into an oval shape, and they were plump with multiple chromocenters and often exhibited one or more prominent eosinophilic nucleoli. The cytoplasm was moderately abundant, slightly granular and lightly eosinophilic (figure 2a right panel).

Concerning the latency time, we observed a doubling of the latency time when the concentration in cells was decreased by 5. We added a comment on this point in the revised manuscript, page 4, line172:

“This latent period was dependent on the final OSCC cell concentration. As illustrated figure 1c, a doubling of the latency time occurred when the cell concentration was decreased by five ( 5x105 cells/mL versus 1x105cells/mL).”

2. Figures 4 and 5: The conclusion that defect in platelet aggregation is due to diminished ADP secretion is not substantiated by the data. The authors should show that addition of ADP restores platelet aggregation induced by low doses of epinephrine or collagen (CRP), two agonists known to be dependent on ADP secretion. If a lack of ADP is involved, aggregation should be restored to a level similar to the one observed with control platelets.

We thank the reviewer for this pertinent comment. We agree with the reviewer that we did not demonstrate that ADP is the only agonist involved in this low platelet’s reactivity. We focused on ADP and thrombin (TRAP) agonists since these molecules are the main platelet agonists involved in the laser injury model. In the revised manuscript

1- We justified the use of ADP and TRAP, page 10, line 371:

“With PRP, we studied platelet aggregation in mice with tongue tumors using two different major activators implicated in our laser injury model as already published and validated by our team, namely thrombin and ADP[24,25,32]. Thrombin and ADP are also the main agonists involved in thrombosis associated with tumor [2].”

2-We moderate our conclusion according to the reviewer comment, page 12 , line 465

Thus, these findings suggest a deficiency of granule storage in mice with tongue tumors resulting in a lower activibility of platelets in bearing lingual tumor mice, independent of the agonist used. This could lead to a lower level of activated aIIbb3 at the platelet surface.”

How do the authors explain that aggregation induced by ADP alone is affected?

One possibility is that P2Y1 get desensitized. This hypothesis is discussed in the revised discussion, page 14, line 535:

“One could postulate that the presence of cancer cells in the bloodstream induced exhausted platelets which can be encountered in cancerous patients even in the absence of consumption coagulopathy. However, we reported that OSCC tumors express high levels of active TF, inducing a TCIPA when present in blood. Alternatively, tumors could secrete ADP[2]. ADP may in turn activates platelets leading to the desensibilization of the P2Y1 receptor and the secretion of the granule contents, leading to exhausted platelets insensitive to ADP. In this case, activated and not resting platelets will be observed by electron microscopy. Also, this hypothesis does not explain the decrease in the reduced level of activated aIIbb3 observed. Another possibility is that OSCC tumors will directly or indirectly act on megakaryocytes leading to the generation of deficient platelets.”

3. An interesting point of this study is the defect of dense granules and whether the altered granule content is the reason behind the abnormality. However, more data to get insights into alpha and dense granules defects should be provided. For example, platelet vWF, fibrinogen and serotonin global content should be quantified by immunoassay/western blot or imaging. Dense granule release (ATP or serotonin release), as well as the α-granule release (PF4 or vWF release; ELISA assay) should be assessed. The number of α-granules should also be evaluated by EM and shown.

We thank the reviewer for this comment. We reanalyzed our electronic microscopy experiments and count the alpha and dense granules. Both are defective. We included theses results in the manuscript page 12, line 465:

“A significant (p<0.0001) decrease in number of dense and alpha granules was found in platelets from mice with tongue tumors (mean of 0,48+/- 0,03 dense granules and 4,4 +/- 0,13 alpha granules per platelet) compared to platelets from tumor-free mice (mean of 1,6 +/- 0,05 dense granules and 5,3 +/- 0,15 alpha granules per platelet) (figure 5b).”

- Discussion: “Our study describes the first case of solid cancer lacking thrombosis risk due to a deficiency in the delta-storage pool.”: As such, the model is not fully supported by data and a more balanced take home message should be provided.

We agree with the reviewer and modified this sentence in our discussion, page 13, line 503:

“Our study indicates that the lack of thrombosis risk in a solid cancer could be due to a deficiency in the storage pool.”

4. Another possible scenario is that carcinoma cells induced exhausted platelets which can be encountered in cancerous patients even in the absence of consumption coagulopathy. Assessing soluble activation markers (sCD40 or GPVI shedding) and the expression level of P-selectin would be valuable. This alternative model should be probed experimentally or at the very least be considered and discussed.

We are grateful to the reviewer for this comment. Indeed, following the counting of alpha granules and dense granules in both mice populations (free mice tumor and tongue tumor mice) we obtained a decreased of alpha and dense granule in tongue tumor mice. We have considered the option of exhausted platelets and modified the discussion, page 14, line 535 (already mentioned point 2):

“One could postulate that the presence of cancer cells in the bloodstream induced exhausted platelets which can be encountered in cancerous patients even in the absence of consumption coagulopathy. However, we reported that OSCC tumors express high levels of active TF, inducing a TCIPA when present in blood. Alternatively, tumors could secrete ADP[2]. ADP may in turn activates platelets leading to the desensibilization of the P2Y1 receptor and the secretion of the granule contents, leading to exhausted platelets insensitive to ADP. In this case, activated and not resting platelets will be observed by electron microscopy. Also, this hypothesis does not explain the decrease in the reduced level of activated aIIbb3 observed. Another possibility is that OSCC tumors will directly or indirectly act on megakaryocytes leading to the generation of deficient platelets.”

Minor comments: 

1. Do the authors have any idea about the amount of cancer cell-derived microparticles in their mice ?

Thanks to the reviewer for this comment. As discussed point 1, these cells do not secrete a substancial quantity of MPs. However, we do not believe MPs are the main focus of our paper and adding such data in the manuscript will add confusion to our main message.

2. Line 308 (and line 319): TRAP is a synthetic peptide, the authors probably meant Thrombin?

We modified this point. The reviewer is right we meant Thrombin.

3. The text size in the figures must be increased.

As asked by the reviewer, the text size of all the figures were increased.

Reviewer 2 Report

Excellent paper

Author Response

we are indebted to the reviewer for this excellent comment.

Reviewer 3 Report

Using AT-84 cell line, Haen et al., analyzed the association between thrombosis risk and OSCC. The authors performed basic aggregometry assays and laser injury model to induce thrombosis. Using these technics, as title indicate the authors concluded that OSCC is associated with low thrombosis risk due to delta-storage pool deficiency in platelets.

First tumor tongue model, as performed is not a model mimicking human OSCC. In cancer biology field, OSCC model closed to human is based to the injection of 4NQQ into the langue of mice. https://www.ncbi.nlm.nih.gov/pmc/articles/PMC7052016/ The authors performed all the studies with only one cell line AT-84. Moc1, Moc2, SCC7, Ca9-22, SAS and CAL27 are good cell lines to use in cancer-associated thrombosis. This cell line AT-84 does not seem a good model, for the induction a tongue tumor.

It is also not clear why delta granules contain less ADP in platelets derived from AT-84-injected mice (tongue tumor) ? the levels of serotonin are changed ?

The title says that cancer is associated with a low risk of thrombosis. Cancer is often associated with high risk of thrombosis, if not associated at all; I do not think that cancer can diminish thrombosis.

Could the author compare tongue tumor group with tongue tumor+apyrase in thrombosis model and also measure size of tumors ?

Discussion does not give any information, why the results are different between in vitro and in vivo model, and why the authors perform these studies and only with one cell line. Title should be differential effects ….. ? What we can conclude from these results in relation to delta-storage pool disease. Is that frequent that OSCC patients have delta storage pool disease and this is associated to a good prognosis, correlation with grade and stage ?

Author Response

Reviewer 3

Using AT-84 cell line, Haen et al., analyzed the association between thrombosis risk and OSCC. The authors performed basic aggregometry assays and laser injury model to induce thrombosis. Using these technics, as title indicate the authors concluded that OSCC is associated with low thrombosis risk due to delta-storage pool deficiency in platelets.

First tumor tongue model, as performed is not a model mimicking human OSCC. In cancer biology field, OSCC model closed to human is based to the injection of 4NQQ into the langue of mice. https://www.ncbi.nlm.nih.gov/pmc/articles/PMC7052016/ The authors performed all the studies with only one cell line AT-84. Moc1, Moc2, SCC7, Ca9-22, SAS and CAL27 are good cell lines to use in cancer-associated thrombosis. This cell line AT-84 does not seem a good model, for the induction a tongue tumor.

We thank the reviewer for these comments and this reference. We developed a 4NQO model in our lab. However, it was not reproductible (meaning the time scale could be really different from mouse to mouse). This is in fact confirm by the review point out by the reviewer (see table number 1).  Moreover, the kinetics of tumors development are different from lab to lab. Thrombosis preclinical models need to be performed in a short period of time and by the same experimenter to be reproductible and accurate. We conclude this 4NQO was not usable in these conditions since the results could be misleading.  

We choose to use AT84 cell, for several reasons. First is it a mouse spontaneous sarcoma lingual cancer cell line described in the literature. Second, this cell line is available (is not the case of the cells line cited by the reviewer). Third, although each mouse model may not fully resume the physiopathology of the human disease, we believe that using an orthotopic syngenic mouse model is better than developing an orthotopic xenogenic mouse model when studying thrombosis associated with cancer. Mice are not immunodeficient and this is an important point since neutrophils/NETs are playing an essential role in the development of immuno-thrombosis.

It is also not clear why delta granules contain less ADP in platelets derived from AT-84-injected mice (tongue tumor)? the levels of serotonin are changed ?

The reviewer is right, we modified the result and discussion sections according to this comment (please see points 3 and 4 of reviewer 1).

The title says that cancer is associated with a low risk of thrombosis. Cancer is often associated with high risk of thrombosis, if not associated at all; I do not think that cancer can diminish thrombosis.

We found this comment unfair and based on not scientific point of view.

Could the author compare tongue tumor group with tongue tumor+apyrase in thrombosis model and also measure size of tumors ?

This point is out of focus to our current publication.

Discussion does not give any information, why the results are different between in vitro and in vivo model, and why the authors perform these studies and only with one cell line. Title should be differential effects ….. ? What we can conclude from these results in relation to delta-storage pool disease. Is that frequent that OSCC patients have delta storage pool disease and this is associated to a good prognosis, correlation with grade and stage ?

We modified the revised discussion and discussed these points page 14, line 427:

“One could postulate that the presence of cancer cells in the bloodstream induced exhausted platelets which can be encountered in cancerous patients even in the absence of consumption coagulopathy. However, we reported that OSCC tumors express high levels of active TF, inducing a TCIPA when present in blood. Alternatively, tumors could secrete ADP[2]. ADP may in turn activates platelets leading to the desensibilization of the P2Y1 receptor and the secretion of the granule contents, leading to exhausted platelets insensitive to ADP. In this case, activated and not resting platelets will be observed by electron microscopy. Also, this hypothesis does not explain the decrease in the reduced level of activated aIIbb3 observed. Another possibility is that OSCC tumors will directly or indirectly act on megakaryocytes leading to the generation of deficient platelets.”

Round 2

Reviewer 1 Report

The authors have addressed my concerns.

Author Response

we are glad to hear that the reviewer one is satisfied with our answers.

Reviewer 3 Report

The discussion of manuscript was improved in this revised version. In future, I would suggest to authors to analyse observed effects also in human OSCC cancer patient biopsies and further investigate molecular mechanisms.

Author Response

We thanks the reviewer for this comment, we will keep it in mind.

This manuscript is a resubmission of an earlier submission. The following is a list of the peer review reports and author responses from that submission.

Round 1

Reviewer 1 Report

The manuscript entitled “Oral squamous cell carcinoma is associated with a low thrombosis risk due to delta-storage pool deficiency” which is contributed by Haen et al. provides an interesting observation that AT-84 OSCC model reducing thrombosis. However, the manuscript is still too preliminary and without solid evidence of molecular mechanism to support their phenomenon observation. Maybe the authors can find other specific cell biology journals or blood vessel system relative journals than Cancers.  

Major:

  1. The OSCC reducing thrombosis phenomenon is not link to any clinical OSCC lethal outcome. Or the authors may validate mice survival rate in their mice model.
  2. Based on my experience, the OSCC tongue injection model will interfere normal tongue function like swallow and cause malnutrition and mice death in very short time. The authors may inject AT-84 cells into buccal than tongue. The buccal injection does not disturb food-intake and easy metastasis cervical lymph node. This may help to find the colt in blood or aberrant pulmonary hemorrhage in malignancy OSCC.
  3. In human OSCC cells, the high malignancy OSCC cells like SAS, are high sticky and easy aggregation through tail vein injection and cause pulmonary venous thromboembolism.

Minor:
1. The nouns are not consistent in the manuscript.  

Reviewer 2 Report

Comments for Author

 In this manuscript, the authors demonstrated that OSCC tumors have a pro-aggregatory phenotype and may be involved in local thrombosis (in the tumor or peritumor area), while there was no elevated systemic thrombosis risk, by using the original mouse model of tongue squamous cell carcinoma. The authors suggested the discrepancy of the thrombotic risk in a pre-clinical mouse model between local tumor environment and the blood stream compartment. The results presented in this manuscript are interesting and may unveil the novel mechanism underlying the thrombosis risk in OSCC cancer patients. There are several concerns as described below.

Comments:

  1. One of the main issues is over-interpretation of their findings. Although the author indeed showed that tongue tumor mice presented the delta-storage pool deficiency in figure 5, there was no experimental evidence to explain the decreased platelet reactivity and the decreased systemic thrombosis risk as of this moment. Thus, I do not believe it appropriate to conclude that OSCC is associated with a low thrombosis risk due to delta-storage pool deficiency.
  2. Over all, this manuscript is not well-organized. The manuscript reveals in many cases lack of details needed to precisely follow the logical lines and understand the experiments. The Results section also includes “Materials and Methods” and “Discussion” section inappropriately. The manuscript has to be extensively reorganized and totally rewritten.
  3. In figure 1c & 1d, the authors used PancO2 cell line showing elevated TF activity and moderate/low TFPI activity to compare with AT-84 cells. Did the authors also perform the experiments using other cancer cell line?
  4. Page3 Line 111: “Fluorescent microscopy (figure 2c and 2d) ~”, Is this correct?
  5. Page3 Line 114: “~tongues without tumor (figure 2e and 2f)” Where is figure 2f?
  6. Page10 Line 276: The author stated “it is the only solid cancer lacking an elevated thrombosis risk” in the Discussion section. Since there is no description about those topics, please provide the references to explain and support the authors’ conclusion and discuss it.
  7. Page10 Line 302~306: From the viewpoint of clinical practice, the potential mechanism existing in humans to counteract the thrombotic potential of OSCC is clinically significant. Please provide more details on the mechanism and discuss these topics in greater depth.
  8. There are numerous typos and small issues with English. The text is grammatically incorrect in some places. The stylistic errors should be corrected.